# Effects of Propolis and Phenolic Acids on Triple-Negative Breast Cancer Cell Lines: Potential Involvement of Epigenetic Mechanisms

**DOI:** 10.3390/molecules25061289

**Published:** 2020-03-12

**Authors:** João Henrique Maia Assumpção, Agnes Alessandra Sekijima Takeda, José Maurício Sforcin, Cláudia Aparecida Rainho

**Affiliations:** 1Department of Chemical and Biological Sciences, Institute of Biosciences, São Paulo State University (UNESP), Botucatu, São Paulo 18618-689, Brazil; joao.maia@unesp.br (J.H.M.A.); jose.m.sforcin@unesp.br (J.M.S.); 2Department of Biophysics and Pharmacology, Institute of Biosciences, São Paulo State University (UNESP), Botucatu, São Paulo 18618-689, Brazil; agnes.takeda@unesp.br

**Keywords:** DNA methylation, DNA methyltransferase inhibitors, EGCG, molecular docking, RASSF1, epigenetic therapy

## Abstract

Triple-negative breast cancer is an aggressive disease frequently associated with resistance to chemotherapy. Evidence supports that small molecules showing DNA methyltransferase inhibitory activity (DNMTi) are important to sensitize cancer cells to cytotoxic agents, in part, by reverting the acquired epigenetic changes associated with the resistance to therapy. The present study aimed to evaluate if chemical compounds derived from propolis could act as epigenetic drugs (epi-drugs). We selected three phenolic acids (caffeic, dihydrocinnamic, and *p*-coumaric) commonly detected in propolis and the (−)-epigallocatechin-3-gallate (EGCG) from green tea, which is a well-known DNA demethylating agent, for further analysis. The treatment with *p*-coumaric acid and EGCG significantly reduced the cell viability of four triple-negative breast cancer cell lines (BT-20, BT-549, MDA-MB-231, and MDA-MB-436). Computational predictions by molecular docking indicated that both chemicals could interact with the MTAse domain of the human DNMT1 and directly compete with its intrinsic inhibitor *S*-Adenosyl-l-homocysteine (SAH). Although the ethanolic extract of propolis (EEP) did not change the global DNA methylation content, by using MS-PCR (Methylation-Specific Polymerase Chain Reaction) we demonstrated that EEP and EGCG were able to partly demethylate the promoter region of *RASSF1A* in BT-549 cells. Also, in vitro treatment with EEP altered the RASSF1 protein expression levels. Our data indicated that some chemical compound present in the EEP has DNMTi activity and can revert the epigenetic silencing of the tumor suppressor *RASSF1A.* These findings suggest that propolis are a promising source for epi-drugs discovery.

## 1. Introduction

Propolis is a resinous mixture produced by honeybees and used in the construction and protection of the hive [1]. This natural origin product is derived from different botanical sources. Thus, the mixed composition of propolis depends on the geographical area and the local flora, which significantly contribute to its heterogeneous and complex chemical composition [2]. It is estimated that raw propolis contains hundreds of chemical compounds, whose extract shows a plethora of biological and pharmacological activities, for instance immunomodulatory [3], antitumoral [4], anti-inflammatory [5,6], antioxidant, and antibacterial, among others [7].

Despite these effects, some mechanisms underlining them are not well-known and pose a great challenge for the scientific community [8]. The propolis extract contains a variety of chemical compounds including flavonoids, terpenes, essential oils and aromatic acids [9]. Several in vitro studies previously demonstrated the cytotoxic effects of propolis extracts as well of isolated specific compounds in cell lines derived from different cancer types such as breast [10,11], colon [12], uterine cervix, and lung [13]. Additionally, it has been demonstrated that propolis can disrupt oncogenic signaling pathways, inhibit cell growth and proliferation, induce apoptosis and anti-angiogenesis [14,15,16,17,18], among other effects [7,9]. Furthermore, propolis or its isolated compounds can also modulate the expression of cancer-related genes such as *TP53* and *CDKN1A* [12] and of proteins such as *MMP2*, *TIMP2* [19], Bcl2, and Bax [20].

Numerous studies reported that chemical compounds obtained from natural sources, such as curcumin from turmeric [21] and flavonoids [22], can inhibit the catalytic reaction mediated by DNA methyltransferases (DNMTs), an enzyme family responsible for establishing and maintaining the DNA methylation patterns in mammal genomes [23,24]. For example, the (−)-epigallocatechin-3-gallate (EGCG) from green tea (*Camellia sinensis)* is a well-known hypomethylating agent. In vitro and in silico evidence demonstrates that EGCG can inhibit the activity of DNA methyltransferases (DNMTs) [22,25].

Recent studies have indicated that chemical compounds present in propolis may target proteins involved in the epigenetic regulation of gene expression. The effects of caffeic acid phenethyl ester (CAPE) on tumor cell growth and survival, angiogenesis and chemoresistance were associated with its action as a histone deacetylase inhibitor (HDACi) in breast cancer cell lines [26]. CAPE also reverted UV-mediated epigenetic modifications in human dermal fibroblasts by directly inhibiting the activity of several histone acetyltransferases (HATi) including p300, CREP-binding protein (CBP), and p300/CBP-associated factor (PCAF) [27]. In addition, it has been demonstrated that caffeic acid inhibited in vitro enzymatic DNA methylation and that the treatment of MCF-7 and MDA-MB-231 breast cancer cells with caffeic or chlorogenic acid partially inhibited the methylation of the promoter region of the *RARB* (retinoic acid receptor beta) gene, suggesting that propolis chemical compounds may function as epigenetic modulators in cancer cells [28].

In this context, the present study was designed to test the hypothesis that the antitumoral effects of propolis may be, in part, mediated by epigenetic mechanisms. Initially, we evaluated the effect of the ethanolic extract of propolis (EEP) on the global DNA methylation content as well as the potential of selected phenolic acids in inhibiting DNA methyltransferases (DNMTs) using in silico and in vitro approaches. Further, we also investigated the potential of EEP and *p*-coumaric acid in inhibiting DNA methylation of the CpG island in the promoter region of the isoform 1A of the gene *RASSF1* (Ras association domain family member 1)*,* as well as its impact in the expression level of the RASSF1 protein. This promoter region was chosen because it is hypermethylated in human cancers [29] and fully methylated in the four breast cancer cell lines selected for our study. The cell lines BT-20, BT-549, MDA-MB-231, and MDA-MB-436 are classified as triple-negative, being characterized by the low expression of progesterone receptor (PR), estrogen receptor (ER), and human epidermal growth factor receptor 2 (HER2). Triple-negative breast cancer is an aggressive disease that is frequently associated with resistance to chemotherapy [30].

## 2. Results

The MTT [3-(4,5-dimethylthiazol-2-yl)-2,5-diphenyltetrazolium bromide] test was used to evaluate the cytotoxic effect of the propolis, phenolic acids, and EGCG in the BT-20, BT-549, MDA-MB-231, and MDA-MB-436 triple-negative breast cancer cell lines. Propolis reduced cell viability in a dose-and-time dependent manner on BT-20 and BT-549 cells. After 72 h of in vitro exposure, the half maximal inhibitory concentration (IC_50_) values of propolis were 18.06 and 25.45 µg/mL for BT-20 and BT-549 cells, respectively. No significant changes on cell viability were observed on MDA-MB-231 and MDA-MB-436 cell lines. While *p*-coumaric acid and EGCG decreased the viability of all cell lines (Figure 1), no effects were detected after in vitro exposure of the same cell lines to caffeic or dihydrocinnamic acids (Appendix A). At the same time point, the IC_50_ for *p*-coumaric acid values were 17.02, 13.94, 22.85, and 23.55 µM; while for EGCG the values were 20.10, 19.16, 24.97, and 18.16 µM for BT-20, BT-549, MDA-MB-231, and MDA-MB-436 cells, respectively.

In order to investigate the potential of propolis as a source of novel DNMT inhibitors (DNMTi), we first investigated the effect of the EEP to change the global content of DNA methylation in the above-mentioned cell lines. After 96 h of in vitro treatment with propolis with a dose of approximately ½ IC_50_ calculated for the sensible cell lines, no differences were detected relative to the respective controls (Figure 2A).

Based on the sensibility of BT-549 cells to *p*-coumaric acid and EGCG, this cell line was selected for further experiments designed to evaluate the potential of *p*-coumaric acid as an epigenetic-drug in comparison to the EGCG, which is a well-known DNMTi. The *RASSF1* locus was chosen because it is frequently hypermethylated in the promoter region of *RASSF1A* alternative transcript. MS-PCR (Methylation Specific-Polymerase Chain Reaction) analysis confirmed that this promoter region is fully methylated in BT-549 cells. In addition, after 96 h of continuous exposure to EEP (10 µg/mL) or EGCG (10 µM) this promoter region was partially demethylated (Figure 2B). No changes in the DNA methylation pattern was observed after the treatment with *p*-coumaric acid. While the treatment with EEP reduced the intracellular levels of the RASSF1 protein, the exposure to *p*-coumaric acid or EGCG does not changed the RASSF1 protein expression levels in this cell line (Figure 2C).

In parallel to the experiments described above, the strategy of molecular docking was used to evaluate the potential interactions of the three phenolic acids with the methyltransferase domain (MTAse) of the human DNMT1 protein. The computational predictions by molecular docking of SAH (*S*-adenosyl-l-homocysteine) and the MTase domain reproduced the molecular model based in the respective co-crystallography. Thus, the molecules of SAH (the endogenous intrinsic inhibitor) and the EGCG (a known DNMTi from natural origin) were used as references. As expected, the docking solution of SAH and EGCG showed the best site occupancy and the greatest number of possible interactions with specific amino acid residues (Figure 3A,B,E) in the catalytic pocket of the protein DNMT1. Although docking solutions were similar among the three phenolic acids, the *p*-coumaric acid was the ligand showing the most common interactions when compared with SAH and EGCG. Table 1 resumes the main results of docking calculations based on the best site occupancy, lower free binding energy and amino acid interactions of each chemical compound.

## 3. Discussion

The antitumoral effects of propolis towards human cancer cell lines has been well documented. The present study aimed to identify natural bioactive molecules derived from propolis that are able to inhibit DNA methyltransferases, leading to the reactivation of silenced genes due to promoter hypermethylation. Thus, four triple-negative breast cancer cell lines and three phenolic acids (i.e., caffeic, dihydrocinnamic, and *p*-coumaric acids) present in the sample of Brazilian propolis were selected for in vitro and in silico analysis.

Our data demonstrated that EEP decreased the viability of the BT-20 and BT-549 cell lines, but this effect was not detected in those of MDA-MB-231 and MDA-MB-436. Unlike Brazilian propolis, Cuban propolis presented a cytotoxic effect on MDA-MB-231 cells [31]. Specific differences in the chemical composition of the propolis samples and the heterogeneity of the genetic and epigenetic background of these cell lines could explain the differential response to propolis treatment in the breast cancer cell lines analyzed in the present study. Among the phenolic acids tested, only the *p*-coumaric acid and the EGCG showed cytotoxic effects in the four triple-negative breast cancer cell lines. The pathways mediating the cytotoxic effects of EGCG have been described in the literature [32,33], while the cytotoxic effects of *p*-coumaric acid in breast cancer cells have been poorly investigated. A short-term preclinical model indicated the involvement of *p*-coumaric acid in the chemoprevention of colon cancer [34]. The potential antitumoral of *p*-coumaric acid has also been demonstrated by the down-regulation and inhibition of EGFR active site in colon cancer cell lines [35,36]. The treatment with this chemical compound induced apoptosis in MCF-7 breast cells in a concentration-dependent manner [37]. Furthermore, this last study demonstrated that the treatment with *p*-coumaric acid was associated with increased acetylation in H3 histone, suggesting its potential for HDAC inhibition [37].

Epigenetic factors, including DNA methylation and histone modifications, work together to regulate essential cellular processes such as developmental programs, genome integrity, gene expression, cell proliferation and survival, and death pathways [38]. Aberrant DNA methylation profiles, including global hypomethylation and gene-specific hypermethylation, contribute to the disruption of epigenetic mechanisms and are considered as a promising field for preventing cancer and therapeutic strategies [39]. The DNMT are a family of enzymes that is responsible for establishing and maintaining the DNA methylation patterns throughout mammalian genomes [40]. The enzymatic methylation reaction consists in the transfer of a methyl group from the substrate S-adenosylmethionine (SAM) to the fifth carbon position of the pyrimidine ring of cytosines located in dinucleotides cytosine–guanine (5′-CpG-3′) [41]. As a result, SAM is converted into SAH. This normal byproduct of methyl donation act as a competitive inhibitor of DNMTs due to its binding in the MTase domain. Besides SAH, DNMT activity can be controlled by small molecules [42]. Therefore, to test the hypothesis that among the complex composition of propolis there are chemical compounds that are able to inhibit DNMTs, we first evaluated the effect of propolis treatment in the global DNA methylation in four breast cancer cells, but no differences were found in the relative methylation content between cells treated and the respective untreated control. Then, we used computational simulations based on docking to evaluate possible interactions between the phenolic acids and the MTase domain of the human DNMT1. Molecular docking is an in silico technique that is widely used in the ligand-protein simulation and has been used to identify new epigenetic inhibitors and to understand the mechanisms of action of known compounds as well as novel drugs for epigenetic therapy [43]. Overall, the docking simulations showed that the analyzed phenolic acids could interact with the MTase domain in a way similar to the intrinsic inhibitor SAH and EGCG, although with higher free binding energy. However, using an in vitro prokaryotic model with the recombinant methylase *M.SssI*, the docking predictions were not validated for *p*-coumaric acid and ECGC.

Based on the molecular docking evidence, we further investigated if the treatment with propolis, *p*-coumaric acid, or EGCG could revert the locus-specific methylation and lead to gene re-expression. The *RASSF1* gene has been considered a target gene for this kind of analysis [44]. This gene has several isoforms, but two of them, *RASSF1A* and *RASSF1C,* have been implicated in cancer origin and progression. These isoforms are transcribed from distinct promoters and each of them has an associated CpG island. However, *RASSF1A* and *RASSF1C* promoter regions show opposite DNA methylation patterns: while the CpG island of *RASSF1A* isoform is frequently hypermethylated in several cancer types, the CpG island of *RASSF1C* remains unmethylated. It has been suggested that the hypermethylation of *RASSF1A* may be a marker for early cancer detection and prognosis [29]. Since several natural compounds present in food and herbs can inhibit DNMT expression and the activity of *RASSF1A*, it has been also considered as a target to demethylating drugs for cancer therapy [44].

Here we demonstrated that EEP and EGCG, but nor *p*-coumaric acid, were able to partly demethylate *RASSF1A* in BT-459 cells. The effect of EGCG on the demethylation of *RASSF1A* or its reactivation has not been previously reported [44]. However, under the experimental conditions used in the present study, demethylation was not associated with an increase in the RASSF1 protein levels. In contrast, although the treatment of BT-459 cells with propolis does not change the methylation pattern of *RASSF1A*, it led instead to a reduced RASSF1 protein expression level. Histone modifications and DNA methylation are key epigenetic events leading to the silencing of *RASSF1A*. It has been suggested that the abrogation of *RASSF1A* can allow *RASSF1C* expression. In a previous study, we analyzed the expression level of these alternative transcripts of *RASSF1* gene by quantitative real time RT-PCR [45]. We described that while the *RASSF1A* transcript is silenced by hypermethylation in breast cancer cell lines, the mRNA of *RASSF1C* is overexpressed in BT-549 cells when compared with epithelial mammary cells [45]. The antibody used in the protein quantification in the ELISA assay is unable to differentiate the 1A and 1C isoforms of the RASSF1 protein, limiting the interpretation of its results. Nevertheless, these data suggest that propolis exposure could reduce the expression of isoform *RASSF1C* or disrupt the ratio RASSF1A/RASSF1C in cancer cells. This finding is relevant because, contrarily to RASSF1A, some studies indicated that RASSF1C has oncogenic properties and could promote cell survival and proliferation [46].

Here, we demonstrate that some component of propolis extract reverted the DNA methylation of an important tumor suppressor gene. New studies are clearly necessary to identify this/these compound(s). Epi-drugs targeting DNA methyltransferases are becoming a promising alternative to improve cancer therapy, since the combined use of DNMTi at low doses might revert resistance to cytotoxic agents, in part, by remove the acquired epigenetic alterations associated with the resistance to therapy [47].

## 4. Materials and Methods

The present study used in vitro and in silico approaches to investigate if propolis-derived molecules can inhibit DNMTs. Detailed study design is given in the Appendix A.

### 4.1. Propolis Sample and Chemical Compounds

The present study used a propolis sample previously collected at the beekeeping section from São Paulo State University (UNESP), Campus of Botucatu, São Paulo, Brazil. Crude propolis was stored at −20 °C and the ethanolic extracts of propolis were (EEP) prepared as indicated by Sforcin et al. [48]. The composition of propolis was previously characterized by Gas Chromatography–Mass Spectrometry (GC-MS) and Thin-Layer Chromatography (TLC). The main constituents of our propolis sample were isolated and identified: flavonoids are present in small quantities in Brazilian propolis (kaempferid, 5,6,7-trihydroxy-3,4′-dimethoxyflavone, aromadendrine-4′-methyl ether); a prenylated *p*-coumaric acid and two benzopyranes: E and Z 2,2-dimethyl-6-carboxyethenyl-8-prenyl-2*H*-benzopyranes); essential oils (spathulenol, (2*Z*,6*E*)-farnesol, benzyl benzoate and prenylated acetophenones); aromatic acids (dihydrocinnamic acid, p-coumaric acid, ferulic acid, caffeic acid, which are common for poplar propolis, 3,5-diprenyl-p-coumaric acid, 2,2-dimethyl-6-carboxy-ethenyl-8-prenyl-2*H*-1-benzo-pyran); di- and triterpenes, among others. Seasonal variations in propolis composition are not significant and are predominantly quantitative [49]. The chemical compounds caffeic, dihydrocinnamic, and *p*-coumaric acids were provided by the Faculty of Pharmaceutical Sciences, São Paulo University (USP–Ribeirão Preto, São Paulo, Brazil), while the (−)-epigallocatechin gallate (EGCG) was purchased from Sigma Aldrich, St. Louis, MO, USA.

### 4.2. Cell Lines and Cell Culture

Four triple-negative breast cancer cell lines (BT-20, BT-549, MDA-MB-231, and MDA-MB-436) were obtained from the Tissue Culture Shared Resource at the Lombardi Comprehensive Cancer Center from Georgetown University, Washington DC, USA. Before the experiments, genomic authentication was conducted and the culture conditions were described previously [50]. High Glucose Dulbecco’s Modified Eagle’s Medium (DMEM, LGC Biotecnologia, Cotia, SP, BR) supplemented with 10% fetal bovine serum (LGC, Biotecnologia, Cotia, SP, BR), 1% of penicillin (10.000 U/mL) and streptomycin (10.000 µg/mL) (Thermo Fisher Scientific, Waltham, MA, USA) was used for all cell lines. For the BT-20 cells, the culture medium was supplemented with Gibco^®^ MEM non-essential amino acids solution (Thermo Fisher Scientific, Waltham, MA, USA).

### 4.3. Cell Viability Assay

Colorimetric MTT assay was performed to assess cell metabolic activity by the ability of mitochondrial NAD(P)H-dependent oxidoreductase enzymes to reduce the soluble yellow tetrazolium salt [3-(4,5)-dimethylthiazol-(-z-y1)-3,5-diphenyltetrazolium bromide] to insoluble purple formazan crystals. Cell lines at exponential in vitro growth were trypsinized with trypsin/EDTA 0.25% solution (LGC Biotecnologia, Cotia, SP, BR). Afterwards, the cells were diluted in 1mL of culture medium, which was counted in the Countess^TM^ Automated Cell Counter (Invitrogen, Carlsbad, CA, USA) and seeded at a density of 2 × 10^3^ cells in 96-well plates. After a period of 24 h for cell adherence, they were exposed to different concentrations of EEP (6.25, 12.5, 25, 50 and 100 µg/mL). Cells were also exposed to the following concentrations of caffeic acid, dihydrocinnamic acid, *p*-coumaric acid, and EGCG: 6.25, 12.5, 25, 50 and 100 µM, during 24, 48, and 72 h. These chemical compounds were either diluted in dimethyl sulfoxide (DMSO) or ethanol (Sigma Aldrich, St. Louis, MO, USA). Untreated control cells exposed to the respective diluents, were used as references. After the treatment, the medium was aspired, and 100 µL of MTT solution (1 mg/mL) was added to each well. Cells were incubated for 4 h at 37 °C. Formazan crystals were dissolved in DMSO (100 µL). The absorbance was measured using the LX800 (BioTeK^®^, Winooski, VT, USA) at 540 nm automated plate reader. Corrected absorbance values were used to estimate the cell viability expressed by the ratio: (A540 average treated cells–A540 average blank)/(A450 average untreated control cells–A540 average blank). The assays were performed in triplicates.

### 4.4. In itro Treatments and DNA Extraction

The breast cell lines were seeded at a density of 1 × 10^5^ cells in 25 cm^2^ culture flasks and incubated at 37 °C. After 24 h, the cells were treated with 10 µg/mL of EEP and 10 µM of either *p*-coumaric acid or EGCG for 96 h. The culture medium was replaced with fresh medium every 24 h. All experiments were performed in triplicates and a mock treatment was done with diluents (control). Then, the cells were allowed to recover for 24 h prior to harvesting. After the treatment, the cells were treated with trypsin, centrifuged and the pellet was frozen at −80 °C. The genomic DNA was obtained by standard proteinase K digestion, followed by phenol/chloroform extraction and ethanol precipitation. Fluorescent DNA quantification was determined with the QuantiFluor^®^ dsDNA Systems and Quantus™ Fluorometer (Promega, Madison, WI, USA), according to the manufacturer’s instructions.

### 4.5. Global DNA Methylation Content

The effect of propolis in the global DNA methylation was investigated with the Imprint Methylated DNA Quantification Kit (Sigma Aldrich, St. Louis, MO, USA), following the manufacturer’s instructions. The methylated DNA fraction was captured using a 5-methylcytosine antibody and was colorimetrically quantified. For each experimental condition, methylation analysis was performed in triplicate (100 ng of input DNA). Three independent biological replicates and fully methylated control DNA were also included in this experiment. The relative content of DNA methylation of propolis treated/control cells was determined by absorbance (A) at 450 nm by the following formula: (A450 average propolis treated cells–A450 average blank)/(A450 average untreated control cells—A450 average blank).

### 4.6. Methylation-Specific Polymerase Chain Reaction of RASSF1A Promoter

Qualitative Methylation Specific-Polymerase Chain Reaction (MS-PCR) was performed to verify the effect of propolis, *p*-coumaric acid, and EGCG treatments in the locus-specific methylation pattern of the breast cancer cell line BT-549. The genomic DNA (1 µg) was modified by sodium bisulfite protocol with EpiTect Bisulfite Kit (Qiagen, Hilden, Germany). After DNA modification, PCR conditions and amplification were conducted as described by a previous study of our group [51].

### 4.7. Expression of the Protein RASSF1

The expression levels of the protein RASSF1 was determined by an enzyme linked immuno-sorbent assay using the human RASSF1 ELISA Kit (Aviva Systems Biology, San Diego, CA, USA). Initially, 1 × 10^5^ cells were exposed to 10 µg/mL of EEP and 10 µM of *p*-coumaric acid or EGCG for 96 h, as described above. Afterwards, the cells were collected by addition of trypsin/EDTA solution 0.25% (LGC Biotecnologia, Cotia, SP, BR), centrifuged, and washed three times in cold PBS 1× (Phosphate-Buffered Saline). The cells were resuspended in PBS 1X, subjected to three freeze/thaw cycles at −20 °C for lysis and centrifuged at 1500× *g* for 10 min at −8 °C to remove cellular debris. The protein concentration in the cell lysates was estimated in the NanoDro 1000 spectrophotometer (Thermo Fisher Scientific, Waltham, MA, USA) and diluted with a standard diluent. ELISA protocol follows the manufacturer’s recommendations. The results were based on the relative optical density (OD) at 450nm (OD450), as follows: (Relative OD450) = (well OD450) – (mean blank well OD450). The standard curve was generated by plotting the mean replicate relative OD450 of each standard serial dilution point versus the respective standard concentration (ranging from 10,000 to 156.25 pg/mL, dilution factor 1/2). The RASSF1 concentrations in the samples were interpolated by linear regression.

### 4.8. Molecular Docking

Docking calculations between ligands and the methyltransferase domain of the human DNMT1 was performed with the AutoDock Vina software [52]. The three-dimensional structure was obtained from The Protein Data Bank (PDBID 4WXX). The chemical structures of probable ligands were retrieved from The PubChem database (pubchem.ncbi.nlm.nih.gov): caffeic acid (CID 689043), hydrocinnamic acid (CID 107), *p*-coumaric acid (CID 637542), and EGCG (CID 65064). SAH (CID 439155) was used as a reference molecule in each step. The area of interest on the MTase domain was defined by establishing a cube at the geometric center of the co-crystalized SAH, with dimensions of 20 × 20 × 20 Å, covering the SAH binding site by employing a grid-point spacing of 1.0 Å. The x, y, and z coordinates for the center MTase domain were −45.55, 61.52, and 6.091, respectively. For each ligand tested, an exhaustiveness of 10 was used. The best docking solution of each ligand was selected based on the lowest free binding energy (Kcal/mol), geometric position and residue contacts analyzed by AutodockTools software [53]. The Root Mean Square Deviation (RMSD) values were calculated according to the default cutoff parameter of AutoDock Vina [52]. For protein surface and image building, we used UCSF Chimera visualization software (University of California, San Francisco, CA, USA) [54].

### 4.9. In Vitro Inhibition of the CpG Methylase M.SssI Assay

The recombinant enzyme *M.SssI* methylates all cytosine residues in double-stranded DNA fragments at CpG dinucleotides. Initially, a fragment of 658 bp from the human gene *MECP2* (chrX:154,030,181-154,030,838) was generated by Polymerase Chain Reaction (PCR). This amplicon contains 26 CpGs and one recognition site of the *BstUI* restriction enzyme (5′-CGCG-3′). The digestion of this PCR product with *BstUI* generates two fragments of 332 bp and 336bp; however, the cleavage is inhibited by cytosine methylation. Thus, the purified fragment of 658bp was used as substrate DNA for the in vitro methylation assay. The methylation reaction contained 400 ng of substrate DNA and 4 U of *M.SssI* methylase (New England Biolabs, Ipswich, MA, USA) in a final volume of 50 µL at 37 °C overnight, as described by Brueckner et al. [55]. *p*-coumaric acid or EGCG were tested at 25, 50, 100, 200, and 400 µM. Positive (without any drug test) and unmethylated (without methylase *M.SssI*) controls were also included in the experiment. After completion, the reaction was inactivated at 65 °C for 15 min, followed by purification and digestion with *BstUI* (50 mM potassium acetate, 20 mM tris-acetate, 10 mM magnesium acetate, and 100 µg/mL BSA, at 60 °C). The visualization of *BstUI* digested fragments on 6% polyacrylamide gel electrophoresis is indicative of unmethylated restriction sites due to the inhibition of the enzyme activity.

### 4.10. Statistical Analysis

The statistical significance of the experimental data compared with untreated controls was determined by a paired t-test or ANOVA, corrected by Dunnett’s test for multiple comparisons. The significance level was 5% and the statistical tests were performed using the GraphPad Prism 8 software (GraphPad Software, Inc., San Diego, CA USA).

## 5. Conclusions

In conclusion, the present study showed that propolis reduced the viability of BT-20 and BT-549 cells, while *p*-coumaric acid and EGCG showed cytotoxic effects in all analyzed triple-negative breast cancer cell lines. Molecular docking simulations indicated that the phenolic acids and EGCG can interact with the MTase domain of the DNMT1 enzyme. Moreover, the potential use of small molecules derived from propolis in the discovery of new epi-drugs is supported by the fact that propolis partially demethylate the promoter region of *RASSF1A* in the BT-549. Further studies are clearly necessary in order to characterize propolis-derived chemical compounds as new epi-drugs.

## Figures and Tables

**Figure 1 molecules-25-01289-f001:**
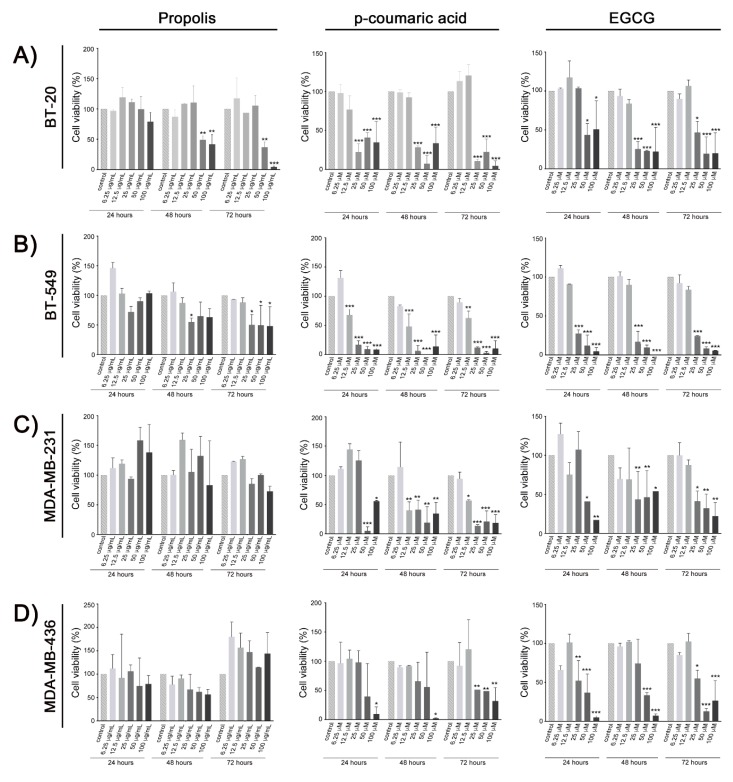
Relative cell viability analysis after in vitro treatment with ethanolic extract of propolis, *p*-coumaric acid, and (−)-epigallocatechin-3-gallate (EGCG) in triple-negative breast cancer cell lines: BT-20 (**A**), BT-549 (**B**), MDA-MB-231 (**C**), and MDA-MB-436 (**D**). Data represent means and standard deviation of three independent experiments. * *p* < 0.05; ** *p* < 0.002; *** *p* < 0.001 in comparison with the untreated controls in the respective period of exposure (24, 48 or 72 h).

**Figure 2 molecules-25-01289-f002:**
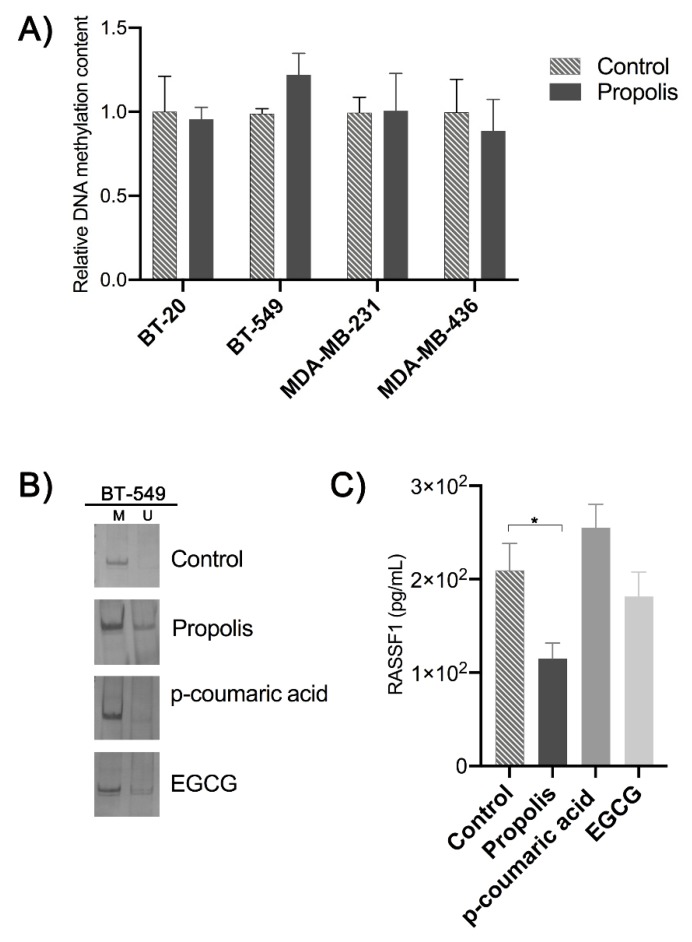
(**A**) Analysis of global DNA methylation content after propolis treatment relative to the respective control. (**B**) Locus-specific DNA methylation analysis by MS-PCR. BT-459 cells are fully methylated at the *RASSF1A* promoter region (M = methylated and U = unmethylated alleles). The in vitro treatment with propolis and EGCG was able to partially demethylate this locus, as evidenced by detection of the PCR product with primer-specific to detect the unmethylated DNA sequence. (**C**) RASSF1 protein expression levels in BT-549 cells after the in vitro treatment with propolis (10 µg/mL), *p*-coumaric acid, and EGCG (10 µM) during 96 h. * *p* < 0.05.

**Figure 3 molecules-25-01289-f003:**
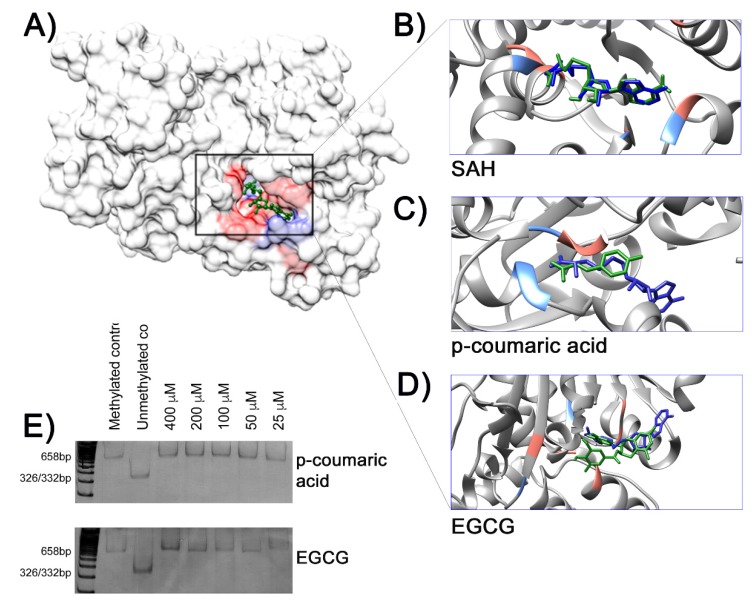
(**A**) Crystalografic model of the methyltransferase (MTase) domain of human DNA methyltransferase 1 (DNMT1) (PDBID:4WXX) complexed with S-adenosyl-l-homocisteine (SAH). The highlighted area in the rectangle indicates the docking of SAH (green), the product of the DNA methylation reaction, in the MTase domain surface model (gray). Hydrophobic contacts between the ligand and amino acids residues are in red, with potential hydrogen bonds shown in light blue. Details of interactions between ligands from docking simulation (green sticks) and amino acid residues are shown in (**B**) SAH, (**C**) *p*-coumaric acid, and (**D**) EGCG. All ligands were overlapped with SAH from the crystallographic model (dark blue sticks). (**E**) In vitro DNA methylation assay. The absence of BstUI restriction fragments in the methylation reactions containing *p*-coumaric acid or EGCG indicates no inhibitory effects of methylase *M.SssI* by these chemical compounds.

**Table 1 molecules-25-01289-t001:** Computational predictions of interactions between ligands and MTase domain of human DNMT1 by molecular docking. Concordant amino acid residues involved in the predicted interactions are indicated in bold.

Ligand	CID	2DMolecularStructures *	Binding Energy(Kcal/mol)	Max RMSD **	Hydrophobic Contacts	Hydrogen Bonds
*S*-adenosyl-homocysteine(SAH)	439155	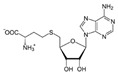	−8.3	4.547	**Phe1145**, **Leu1151**, Glu1168, Cys1191, Leu1247,**Ala1579**, Val1580	Met1169, Asp1190, **Ans1578**
caffeic acid	689043	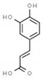	−6.7	6.809	Ser1146, Gly1147, Cys1148, **Asn1578**, **Ala1579**	Gly1149, Gly1150, **Leu1151**, **Val1580**
hydrocinnamicacid	107	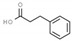	−5.2	7.170	Ser1146, Gly1147, **Asn1578**, Gly1223	**Leu1151**, **Val1580**
*p*-coumaric acid	637542	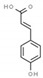	−6.0	6.049	Gly1147, **Asn1578**, **Ala1579**	Gly1149, Gly1150, **Leu1151**, **Val1580**
(−)-epigallocatchin−3-gallate	65064	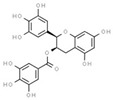	−10.4	0.1564	Arg1312, **Asn1578**, Val1580, Gly1223,Gly1147, **Phe1145**	Glu1266, Arg1310

CID–PubChem Compound ID number; (*) Retrieved from ChemSpider (https://www.chemspider.com/Default.aspx); (**) RMSD-Root-Mean-Square Deviation.

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
