# Peer review of "Effects of Propolis and Phenolic Acids on Triple-Negative Breast Cancer Cell Lines: Potential Involvement of Epigenetic Mechanisms"

_molecules, 2020, doi:10.3390/molecules25061289_

Round 1

Reviewer 1 Report

The article “Effects of propolis and phenolic acids on triple-negative breast cancer cell lines: potential involvement of epigenetic mechanisms” by Assumpção et al. describes in vitro and in silico studies on the effects of propolis and phenolic acids on select triple-negative breast cancer cell lines. Overall the article presents some novelty and is well written. Below, I am addressing some minor comments for authors in order to improve quality of their work.

  1. Results section, line 91, the authors are listing obtained IC50 values. Please provide for which time point those values are reported. Also, in line 92 should be 23.55 instead of 23,55.
  2. Figure 1, usually cell viability is presented as a percent of growth inhibition and a scale from 0-100% is used. Please explain the labeling on “y axis”, which is 0-2 without unit or %.
  3. Line 114, wrong tense is used, there is “evaluated” should be “evaluate”, please correct.
  4. Line 123, should be Figure 2C instead of 1C. Please correct.
  5. Materials and methods, please provide information about compound used as a positive control in cell viability assay.

Author Response

The article “Effects of propolis and phenolic acids on triple-negative breast cancer cell lines: potential involvement of epigenetic mechanisms” by Assumpção et al. describes in vitro and in silico studies on the effects of propolis and phenolic acids on select triple-negative breast cancer cell lines. Overall the article presents some novelty and is well written. Below, I am addressing some minor comments for authors in order to improve quality of their work.

Dear Reviewer,

Thanks for your efforts in reviewing our manuscript. The constructive suggestions improved the quality of this work. We have included a workflow as supplementary material to adequately described the strategy of the present study and the methods used to test our hypothesis. 

The following are our point-by-point responses to the comments:

  1. Results section, line 91, the authors are listing obtained IC50 values. Please provide for which time point those values are reported. Also, in line 92 should be 23.55 instead of 23,55.

 Response: These suggestions were accepted and included in the revised version of our manuscript. Each modification made in the text was indicated as a gray highlight.

  1. Figure 1, usually cell viability is presented as a percent of growth inhibition and a scale from 0-100% is used. Please explain the labeling on “y axis”, which is 0-2 without unit or %.

Response:  We have reported the results of MTT assay as relative cell viability, i.e., the normalized fraction of live cells obtained by dividing the value of each treatment by the mean value of the control samples (1.0 or 100%). We agree that, usually, cell viability is presented as percentages. Thus, we have transformed our results in this scale in figure 1 and supplementary figure 1.  

  1. Line 114, wrong tense is used, there is “evaluated” should be “evaluate”, please correct.

 Response: The suggestion was accepted and included in the revised version of our manuscript.

  1. Line 123, should be Figure 2C instead of 1C. Please correct.

Response: The suggestion was accepted and included in the revised version of our manuscript

  1. Materials and methods, please provide information about compound used as a positive control in cell viability assay.

Response: Given that cytotoxic effects of propolis is well documented in several human cancer cell lines, including breast cancer cell lines, we have not included a positive control in the cell viability assays. The treatment in vitro with the ethanolic extract of propolis (EEP), caffeic, dihydrocinnamic, and p-coumaric acids as well as EGCG was performed to determine the IC50 values and to drive the choice of the better concentration of the drugs to be used in the following assays (in vitro treatment before the analysis of DNA methylation content, MS-PCR, and protein expression). Since well-known epi-drugs revert DNA methylation at lower doses but induce cytotoxic effects at high doses, we selected the ½ IC50 to test our hypothesis that these phenolic acids or other components present in the EEP could inhibit the DNA methylation catalyzed by the enzyme DNMT. Further, since the EEP is characterized as a complex mixture with no major chemical compounds, we expected that propolis-derived chemicals are present at low quantities in the EEP.

Reviewer 2 Report

The authors provide the study showing anti-cancer activity of propolis, that may be connected with inhibition of DMTS.

The manuscript is well written, study well design but still there are important issues that need to be addressed.

Major

  1. The anticancer activity of propolis extract was already expensively studied, also in the context of triple negative breast cancer. Thus the only novel finding is connection it anticancer activity with inhibition of DMTS.

The novelty of the findings should be more underlined and better documented

  1. Despite the propolis extract was most effective in assays on Figure2B and C. Its anticancer activity was rather mild in comparison to reference compounds. It shows that there is no simple connection between inhibition of DNA methylation and anticancer activity. Note, that cytotoxicity of Decytabine its limited, especially in low concentrations regarded as more „anti-epigenetic”.

Unfortunately, these results are the only showing a great activity of propolis exstract. There are obtained on single cell line. Additionally, the other assay related to DNA methylation was performed only on reference compounds. It would be great to provide the data supporting anti-methylation activity of propolis extract in vitro. As assay turn out to be negative positive control need to be included (Decytabine?)

  1. Moreover despite such promising result, the bioinformatic analysis was performed only for reference compounds but not the other compounds present in the ethanol even in order to propose the chemicals worth further studies. Such analysis will allow better interpretation of the results.

Minor

  1. Authors underlined variability of the propolis with is natural product. It means that the composition of actual preparation used in the study should be provided.
  2. The presentation of MTT data must be improved. The names of the cell lines as well as concentrations values should appear on the graphs.

Author Response

The authors provide the study showing anti-cancer activity of propolis, that may be connected with inhibition of DNMTs.

The manuscript is well written, study well design but still there are important issues that need to be addressed.

Dear Reviewer,

Thanks for your efforts in reviewing our manuscript. The constructive suggestions improved the quality of this work. We have included a workflow as supplementary material to adequately described the strategy of the present study and the methods used to test our hypothesis. 

The following are our point-by-point responses to the comments:

Major

  1. The anticancer activity of propolis extract was already expensively studied, also in the context of triple negative breast cancer. Thus, the only novel finding is connection it anticancer activity with inhibition of DNMTs. The novelty of the findings should be more underlined and better documented.

 Response: Thank you for your helpful suggestions.  We agree that the anticancer activity of propolis has been extensively reported in several cell lines derived from different types of human cancer. In the present study, we hypothesized that these effects of propolis in cancer cell lines are also mediated by epigenetic mechanisms. If so, propolis should be a promising new natural resource for epi-drug discovery. Indeed, our goal was to evaluate if small molecules present in the propolis composition could be able to interact with human DNMT1 using an in-silico approach. Further, we conduct some experimental assays in TNBC cell lines to investigate the DNA methylation changes after in vitro exposure to propolis and selected phenolic acids detected in the propolis sample.  Although we not identified the DNMTi present in the EEP, our results demonstrate that propolis reverted the DNA methylation of an important tumor suppressor gene.  We believe that our manuscript has an impact in the field because epigenetic drugs targeting DNA methyltransferases are becoming a promising alternative to improve cancer therapy and to reverse acquired therapy resistance.

  1. Despite the propolis extract was most effective in assays on Figure2B and C. Its anticancer activity was rather mild in comparison to reference compounds. It shows that there is no simple connection between inhibition of DNA methylation and anticancer activity. Note, that cytotoxicity of Decytabine its limited, especially in low concentrations regarded as more, anti-epigenetic”  Unfortunately, these results are the only showing a great activity of propolis extract. There are obtained on single cell line. Additionally, the other assay related to DNA methylation was performed only on reference compounds. It would be great to provide the data supporting anti-methylation activity of propolis extract in vitro. As assay turn out to be negative positive control need to be included (Decytabine?)

Response: We appreciate the reviewer for arising this important question. Evidence suggested that use of DNMT inhibitors such as decitabine, at low doses, are able to revert DNA methylation pattern of important epigenetically silenced genes in cancer and might reverse resistance to cytotoxic agents. To the best of our knowledge, the present study shows, at first time, that propolis was able to revert the RASSF1A hypermethylation in TNBC (figure 2).

While at high doses decitabine results in cell death, at low doses, it induces global hypomethylation of the genome. DNMT inhibitors are classified into two groups:  the nucleotide analogues, such as decitabine and azacytidine, and the non-nucleotide analogues. The analogues of nucleotides are incorporated in the newly synthesized DNA chains during replication and can covalently trap DNA methyltransferases, leading to depletion of active enzymes. In this context, decitabine was not used in our experiments because propolis-derived compounds are non-nucleotide homologous. Thus, it is expected that these molecules exert their DNA methylation inhibitory activity by different mechanisms, mainly by directly interacting with the MTase domain of DNMTs and competing with its intrinsic inhibitor. Future studies of our research group will include new assays to better address the DNMTi activity of propolis-derived molecules.

  1. Moreover, despite such promising result, the bioinformatic analysis was performed only for reference compounds but not the other compounds present in the ethanol even in order to propose the chemicals worth further studies. Such, analysis will allow better interpretation of the results.

Response: Currently, we are conducting new bioinformatics analysis and developing a pipeline for molecular screening of new DNMTi derived from propolis extracts. To cover the complexity and heterogeneity of the chemical composition of propolis, we have screened abstracts on PubMed and then the full text of selected articles has been manually reviewed to create a list of chemicals compounds from propolis (approximately 1,000 compounds were retrieved). This wide screening and computational prediction based on molecular docking will indicate the most promisor DNMTi. The new candidates will be validated by assays of DNMT activity in our ongoing studies. Thus, the large-scale screening for other potential DNMTi in the propolis is out of the scope of the present manuscript.

Minor

  1. Authors underlined variability of the propolis with is natural product. It means that the composition of actual preparation used in the study should be provided.

Response: As suggested, we included a description of the propolis composition used in the present study. Please, see the Materials and Methods section (item 4.1. Propolis sample and chemical compounds).

  1. The presentation of MTT data must be improved. The names of the cell lines as well as concentrations values should appear on the graphs.

Response: These suggestions were accepted and included in the revised version of figures showing the graphs of cell viability results.

Round 2

Reviewer 2 Report

I have no further comments